# Evaluation of continuous quality improvement of tuberculosis and HIV diagnostic services in Amhara Public Health Institute, Ethiopia

**Melashu Balew Shiferaw**[ID][1]*, **Abay Sisay Misganaw**[2]

**1** Research and Technology Transfer Directorate, Amhara Public Health Institute, Bahir Dar, Amhara, Ethiopia, **2** Department of Medical Laboratory Sciences, College of Health Sciences, Addis Ababa University, Addis Ababa, Ethiopia

* bmelashu@gmail.com

## Abstract

### Background

Unreliable laboratory results lead to unnecessary tests, procedures or treatments which may harm the patient. Continuous quality improvement (CQI) is a useful objective tool to improve processes and services. The use of quality indicators that meet requirements for effectiveness is an important quality improvement tool. However, the quality of critical aspects of pre-examination, examination, and post-examination processes have not been evaluated in Ethiopia including our setting. Hence, this study aimed to assess the performance of continuous quality improvement of TB and HIV laboratory tests in the Amhara Public Health Institute (APHI).

### Methods

A cross-sectional study was conducted to evaluate the quality indicators of advanced TB and HIV related laboratory tests in APHI from 01 January to 30 September 2019. HIV viral load, exposed infant diagnosis (EID), GeneXpert and TB culture quality indicators data were used as a quality improvement tool and evaluated in comparison to established targets. Data were extracted from excel database and record review of patient information, and entered and analyzed using SPSS V20 software.

### Results

A total of 26,487 samples were received from 01 January to 30 September 2019. The overall specimen rejection rate was 0.43% (115/26,487). Specifically, viral load and TB culture had 0.43% and 1.14% rejection rates, respectively. The highest monthly rejection was documented for TB culture (5.3%) and viral load (2.4%) in September 2019. Centrifugation problems (46.1% [53/115]) and the use of the wrong container (40.9% [47/115]) were the main reasons for the rejections. Moreover, EID test was interrupted for a total of 54 days and 22 days due to reagent stock out and equipment down time, respectively. Similarly, about 82%

**Data Availability Statement:** All relevant data are within the manuscript.

**Funding:** The author(s) received no specific funding for this work.

**Competing interests:** The authors have declared that no competing interests exist.

of viral load and 100% of the EID tests had long turnaround time (TAT) with an average of 24.1 and 29.3 days respectively in September 2019.

## Conclusions

There were high rates of TB culture and viral load specimen rejection, and EID test interruptions. The TAT of viral load and EID tests were longer than the targeted goal (10 days) average TAT. Hence, training of sample collectors, functional equipment maintenance systems and supply chain management are recommended for continuous quality improvement.

## Introduction

Quality indicators are useful objective tools to improve processes and services. In modern clinical medicine, laboratory tests play an important role in diagnosing, monitoring, and evaluating patient outcomes. Hence, the implementation of performance measurements to evaluate the pre-analytical, analytical and post-analytical stages of the total testing process is therefore needed to maximize the overall testing cycle and the quality of patient care [1].

Pre-analytical error and post-analytical error data are well documented. Occurrence of the errors at the pre- and post-analytical phases currently appears to be more vulnerable than the analytical phase [2]. Studies revealed that 46%–68.2% of laboratory errors predominated in the pre- analytical phase, and 18.5%–47% errors recorded in the post-analytical phase of the laboratory testing process [3–6].

The use of quality indicators that meet requirements for effectiveness and harmonization is an important quality improvement tool [7]. It can measure how well an organization meets the needs and requirements of users and the quality of all operational processes. Monitoring quality indicators in daily work can reduce laboratory errors and risk to patient safety by identifying problems in all phases of the laboratory process, allowing their correction [8].

Stock out of reagents, equipment down time, and sample rejection rates are common pre-analytical phase quality indicators, whereas the performance of proficiency testing (PT) in external quality assessment (EQA) program, internal quality control (IQC) and contamination or error rates are analytical phase indicators. Minimum recommended quality indicators for the post-analytical phase are turnaround time (TAT), the percentage of incorrect laboratory test reports, and notification of critical results [8, 9].

Laboratories should deliver accurate, reliable and timely results to customers. When the results are compromised in quality and/ or delayed, it could have an impact on patient management. As a result, the clinician may interpret the results as actionable which, in turn, can lead to unnecessary tests, procedures or treatments which may result in patient harm [10].

According to the World Health Organization (WHO), drug resistance TB and HIV are global challenges that need quality-assured laboratory tests for better treatment options and control of the burden [11]. In addition to the drug resistance burden, if there is poor quality laboratory service, the problem will be magnified.

In Ethiopia, where the estimated proportion of TB cases with multidrug resistant/ rifampicin-resistant tuberculosis (MDR/RR-TB) was16% among previously treated cases and reported a total of 428,472 HIV patients receiving antiretroviral therapy in 2018 [12, 13], laboratory tests are performed to control drug resistance and monitor disease progression. TB culture is performed to identify treatment failures, exposed infant diagnosis (EID) to identify HIV status in exposed infants, and viral load to monitor confirmed viral failure and estimate disease progression. However, the quality of critical aspects of pre-examination, examination and post-

examination processes of TB and HIV related laboratory services have not been evaluated. Hence, this study aimed to use quality indicators to assess the performance of these four important tests, using nine months quality indicators data in the Amhara Public Health Institute (APHI).

## Materials and methods

### Study design

A cross-sectional study was conducted to evaluate the continuous quality improvement of advanced TB and HIV related laboratory tests in APHI from 01 January to 30 September 2019.

### Study setting

APHI is a government public health institute located in Bahir Dar town, Ethiopia at 11˚60′N latitude and 37˚37′E longitude. The institute has three main directorates (laboratory, public health emergency management, and research and technology transfer). It provides laboratory diagnostic services including TB (culture for treatment failure, and GeneXpert for rifampicin resistance), EID and viral load diagnosis to identify HIV status of infants and confirm antiretroviral failure, respectively. These TB and HIV related tests are requested from referring peripheral health facilities in Amhara region through a sample referral network. APHI is used as a reference testing center for more than 150 health facilities. All the laboratory tests investigated in APHI including HIV viral load, EID, TB culture and GeneXpert have been accredited by the Ethiopian National Accreditation Office (ENAO) in complying with the ISO 15189:2012 standard since October 2018 [14].

### Data collection, and quality indicators and objective of each

In APHI, quality indicators were established to monitor the laboratory performance. EQA PT, contamination rate, error rate specimen rejection rate, TAT, IQC performance, stock out, service interruption and equipment down time were the APHI quality indicators (Table 1). Each target was established by reviewing a minimum of 3 months of data at APHI and also reviewing international literature.

All of the requested MDR TB, EID and viral load requested tests were included consecutively in the study. Data were collected using a record review of monthly quality indicator reports from January to September 2019. The established quality indicators at the pre-

**Table 1. Quality indicators established in APHI, 01 January to 30 September 2019.**

| SN | Quality indicators | Target |
|----|--------------------|--------|
| 1. | Equipment down time | Maintained within 5 days |
| 2. | Stock outs | 0% |
| 3. | Specimen rejection | Less than 2% |
| 4. | EQA PT performance | ≥80% |
| 5. | TAT | 90% of tests within defined TAT |
| 6. | IQC performance | 100% |
| 7. | Contamination rate | <5% for TB culture |
| 8. | Error rate | <5% for GeneXpert |
| 9. | Service interruption | 0% |

EQA: external quality assessment; IQC: internal quality control; PT: proficiency testing; SN: serial number; TAT: turnaround time; TB: tuberculosis.

examination phase were equipment down time, stock out, test interruption, and specimen rejection rate.

Equipment down time was set as a quality indicator of APHI virology and bacteriology reference laboratories. All of the equipment should be repaired within 5 days if there is a failure. The equipment maintenance records for each of the following: Abbott 2000SP and Abbott 2000RT (used for extraction and detection of ribonucleic acid (RNA) of EID and HIV viral load); GeneXpert analyzer; and 35˚C-37˚C incubators (used for TB culture). No reagent stock out was set as a quality indicator by APHI. Reagent inventory records such as bin card, internal facility request and resupply form (IFRR), and report and requisition form (RRF) were reviewed to confirm the occurrence of stock outages.

An Excel database maintained in the central reception was used to review specimen rejections. Sputum for drug resistant tuberculosis (DRTB) /MDRTB, dried blood spot for EID and blood plasma for viral load samples were submitted at APHI central reception. The entire submitted patient samples were evaluated for sample quality based on pre-set sample rejection criteria. Sputum samples were rejected when there was labeling error, leaking specimen container, sample volume less than 2ml, sample received 5+ days after collection, use of the wrong container, specimen containing blood and food remnants or exposure to temperature exceeding 8˚C. The plasma sample was rejected if samples were collected using the wrong container, not-centrifuged, hemolyzed, old (if it was delayed >5 days after collection when transported to APHI at 2–8˚C), insufficient volume, labeling error and if not transported below 8˚C temperature. Dried blood spots (DBS) were rejected when there was hemolysis, insufficient volume of spots, clotting or labeling errors. Based on the APHI policy, rejected samples should be communicated to customers and another sample should be recollected for the requested test. The targeted specimen rejection rate was less than 2%.

Proficiency testing performance, contamination rate, error rates, and internal quality control were used for examination phase quality indicators. All of the evaluated tests (TB culture, GeneXpert, EID, and viral load) participated in proficiency testing from Oneworld Accuracy, which is an approved EQA PT provider. Panels of samples prepared by the PT provider were sent to APHI. In APHI, the PTs were analyzed as patient samples and results were submitted online on the Oneworld Accuracy website. Proficiency testing performance of 80% and above was the target goal [15]. Less than 5% target was set as the target goal for the contamination rate and error rate quality indicators by APHI. Applicable test records were reviewed to evaluate examination phase performances.

Moreover, turnaround time was a post-examination phase quality indicator that 90% of tests should be released within defined TAT: viral load = 10 days; EID = 10 days; TB negative culture = 48 days; TB positive culture = 64 days; and GeneXpert = 2 days. The TAT was calculated from login and log out of records of each test achieved at the excel database of the central reception. The TAT of each test was evaluated based on established targets of the institute (Table 1).

## Data review, cleaning and data analysis

The data were entered and analyzed using SPSS V20 software. Selected quality indicators were compared with the defined target and rated as good if complied or not good if not compliant with the established target.

## Ethical consideration

The data were not collected from patients directly since the specimens were collected from the peripheral health facilities and sent to the testing center through a referral network. Before

data collection, ethics approval and official permission (Reference number: HRTT/03/139/2018) was obtained from the APHI research and technology transfer directorate to use patient records from the APHI excel database, quality indicator reports and record review of different quality related records in the laboratory. The Research and Technology Transfer Directorate waived the requirement for informed consent.

## Results

### Test statistics and specimen rejection

In Amhara Public Health Institute, a total of 26,487 HIV and TB related test requests with samples were received from 01 January to 30 September 2019. Of which, 26372 (99.6%) had acceptable quality. Among the acceptable samples, 24517 (93.0%) were blood plasma samples for viral load, 862 (3.3%) were dried blood spot samples for EID, 694 (2.6%) were sputum samples for TB culture and 299 (1.1%) were sputum samples for TB GeneXpert tests (Table 2).

A total of 115 were rejected from January to September 2019. The overall specimen rejection rate was 0.43% (115/26,487). Specifically, viral load, TB culture and TB GeneXpert specimens had 0.43% (106/24623), 1.14% (8/702) and 0.33% (1/300) rejection rates, respectively. High monthly sample rejection was documented for TB culture and viral load in September 2019 with 5.3% (5/95) and 2.4% (61/2557) rejection rates, respectively (Table 3).

Centrifugation problems (46.1% [53/115]) and the use of the wrong container (40.9% [47/115]) were the main reasons for specimen rejection. Viral load specimen rejections were due to centrifugation problems (53/106), use of wrong container (47/106), insufficient volume (5/106) and hemolysis (1/106). Old samples (5/8), use of a wrong container (1/8), insufficient volume (1/8) and uncontrolled temperature (1/8) were causes of rejection for TB culture samples. A single GeneXpert sample was rejected because of insufficient volume. Rejections of specimens were consistently communicated to submitting clinic. However, there was no evidence of receipt of another sample for the requested tests.

### Test interruption

HIV viral load and GeneXpert testing did not experience service interruptions (Table 4). TB culture and EID testing was repeatedly interrupted. Mycobacteria growth indicator tubes were frequently unavailable. Even if a back-up laboratory was assigned, the stock out was a national problem and only the Lowenstein-Jensen culture method was available. EID testing was interrupted by both reagent stock out (HIV qualitative control, 2 months) and equipment downtime (Abbott2000RT, 7 days, March 2019; Abbott2000SP, 15 days, September 2019).

### Genexpert error rates and TB culture contamination rates

In the Amhara Public Health Institute, the error rate was used as a quality indicator to follow TB GeneXpert tests and contamination rate was used to follow TB culture. Both of the indicators were targeted to be less than 5%. Interestingly, all of the monthly error rates and contamination rates reported were within the targeted limit that had been established by the institute. TB culture contamination rate increased from March (2.8%) to June (5.0%) in 2019. Then, it was continuously decreased up to the end of August 2019 (1.0%). Regarding TB GeneXpert, the average error rate over the study period was 1.4%, with monthly error rates ranging 0% - 4.0% (Figs 1 and 2).

**Table 2. Monthly test statistics, APHI, 01 January to 30 September 2019.**

| Tests | 2019 | | | | | | | | | | |
|---|---|---|---|---|---|---|---|---|---|---|---|
| | Jan | Feb | Mar | Apr | May | Jun | Jul | Aug | Sep | Total | |
| | | | | | | | | | | Done | Rejected |
| Viral load | 2571 | 2710 | 3545 | 2018 | 2903 | 2304 | 3340 | 2630 | 2496 | 24517 | 106 |
| EID | 85 | 179 | 0 | 0 | 271 | 87 | 85 | 90 | 65 | 862 | 0 |
| TB culture | 77 | 69 | 87 | 77 | 91 | 32 | 122 | 54 | 85 | 694 | 8 |
| GeneXpert TB | 4 | 7 | 9 | 60 | 47 | 25 | 56 | 45 | 46 | 299 | 1 |
| Total | 2737 | 2965 | 3641 | 2155 | 3312 | 2448 | 3603 | 2819 | 2692 | 26,372 | 115 |

EID: exposed infant diagnosis; TB: tuberculosis.

## Performance of proficiency testing

The laboratories participated in April 2019 in the international PT from Oneworld Accuracy, which is an approved PT provider. All of the assays in the PT program met the minimum requirement for quality performance. Specifically, viral load, EID and GeneXpert had 100% performance as evaluated in April 2019. TB culture PT performance was 80%, which fulfilled the minimum passing mark.

## Turnaround times

The majority of the tests conducted for TB related laboratory diagnosis had acceptable TAT. TB culture tests were released within the targeted TAT except in September 2019. Minimum and maximum average TAT was 43.3 and 65.7 days documented in February and August 2019, respectively. TB GeneXpert tests were consistently completed within the target TAT in March, June and July 2019. However, the average TAT across the study period was 2.3 days with a range of 1.0 to 4.0 days.

Regarding the viral load tests, the average TAT over the study period was 16.5 days, ranging 7.6 to 24.1 days. The TAT improved from January to July 2019 except May 2019 with 18.6% of requested viral load tests out of the defined TAT. Furthermore, the TAT was longer in August and in September 2019 12.6%, and 81.5% of viral load tests were delayed, respectively. EID tests were not done in January, March, and April. Approximately, 35% of the EID tests were delayed in May 2019 with increasing delays occurring throughout the remainder of the study period. During the 9 month reporting period, the EID had a minimum TAT of 23.4 days (February 2019) and maximum TAT of 45.6 days (May 2019) (Tables 5 and 6).

**Table 3. Specimen rejection rates of tests at APHI reference laboratories, January to September 2019.**

| Tests | 2019 | | | | | | | | | |
|---|---|---|---|---|---|---|---|---|---|---|
| | Jan | Feb | Mar | Apr | May | Jun | Jul | Aug | Sept | Total |
| Viral load | 9(0.35%) | 0 | 5(0.1%) | 0 | 1(0.03%) | 0 | 5 (0.15%) | 25(0.94%) | 61 (2.4%) | 106 (0.43%) |
| EID | 0 | 0 | ND | ND | 0 | 0 | 0 | 0 | 0 | 0 |
| TB culture | 1(1.3%) | 2(2.8%) | 0 | 0 | 0 | 0 | 0 | 0 | 5 (5.5%) | 8(1.14%) |
| GeneXpert | 0 | 0 | 0 | 0 | 0 | 0 | 0 | 0 | 1(2.1%) | 1(0.33%) |

EID: exposed infant diagnosis; TB: tuberculosis; ND: not done.

**Table 4. Stock out, equipment down time, and service interruption days APHI, January to September 2019.**

| Tests | 2019 | | | | | | | | |
|---|---|---|---|---|---|---|---|---|---|
| | Jan | Feb | Mar | Apr | May | Jun | Jul | Aug | Sept |
| Viral load | 0 | 0 | 0 | 0 | 0 | 0 | 0 | 0 | 0 |
| EID | 0 | 0 | Yes/26 days | Yes/28 days | 0 | 0 | 0 | 0 | Yes/15 days |
| TB culture | 0 | 0 | 0 | MGIT yes/15 | MGIT yes/31 | MGIT yes/31 | MGIT yes/31 | MGIT yes/31 | MGIT yes/31 |
| GeneXpert TB | 0 | 0 | 0 | 0 | 0 | 0 | 0 | 0 | 0 |

EID: exposed infant diagnosis; MGIT: Mycobacteria growth indicator tube; TB: tuberculosis.

## Discussion

In this study, the overall specimen rejection rate was 0.43%. Specifically, it was 1.14% for TB culture and 0.43% for viral load tests from January to September 2019. Surprisingly, high monthly sample rejection was documented for TB culture and viral load in September 2019 with 5.3% and 2.4% rejection rates respectively. Those findings were higher compared to the 2% monthly quality indicator target established by the institute. Also, the finding is higher than the specimen rejection of 0.26% reported by Cao et al in Huston, United States [16]. This leads to a high rate of repeat specimen collection, delay in result availability and high rate of specimen/ test abandonment [17, 18]. It compromise patient safety, waste resources, patient discomfort, and potential patient complications especially if there are critical values [19–21]. There was no evidence of repeat collection of another sample for the requested tests. Hence, the laboratory should proactively take actions such as training of sample collectors on how to

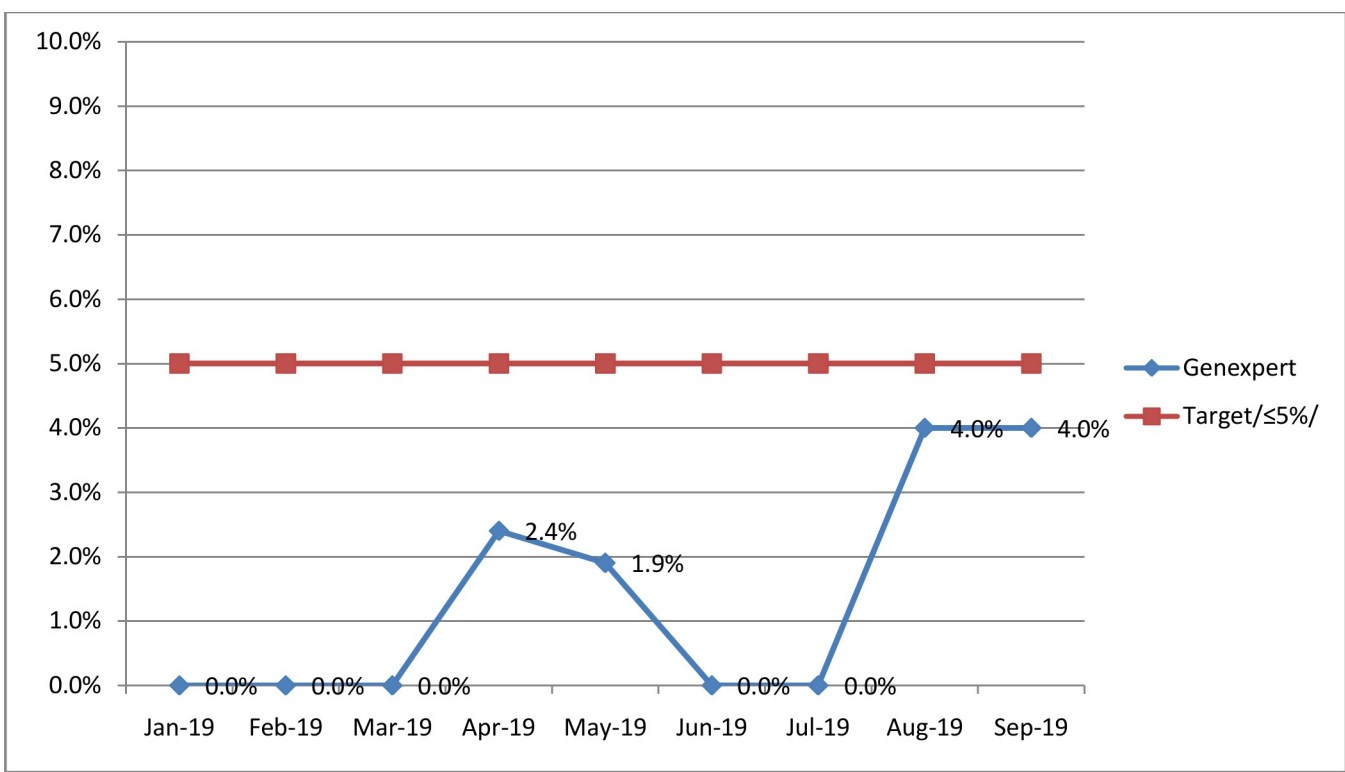

**Fig 1. Trend of GeneXpert error rates in APHI from January to September 2019.**

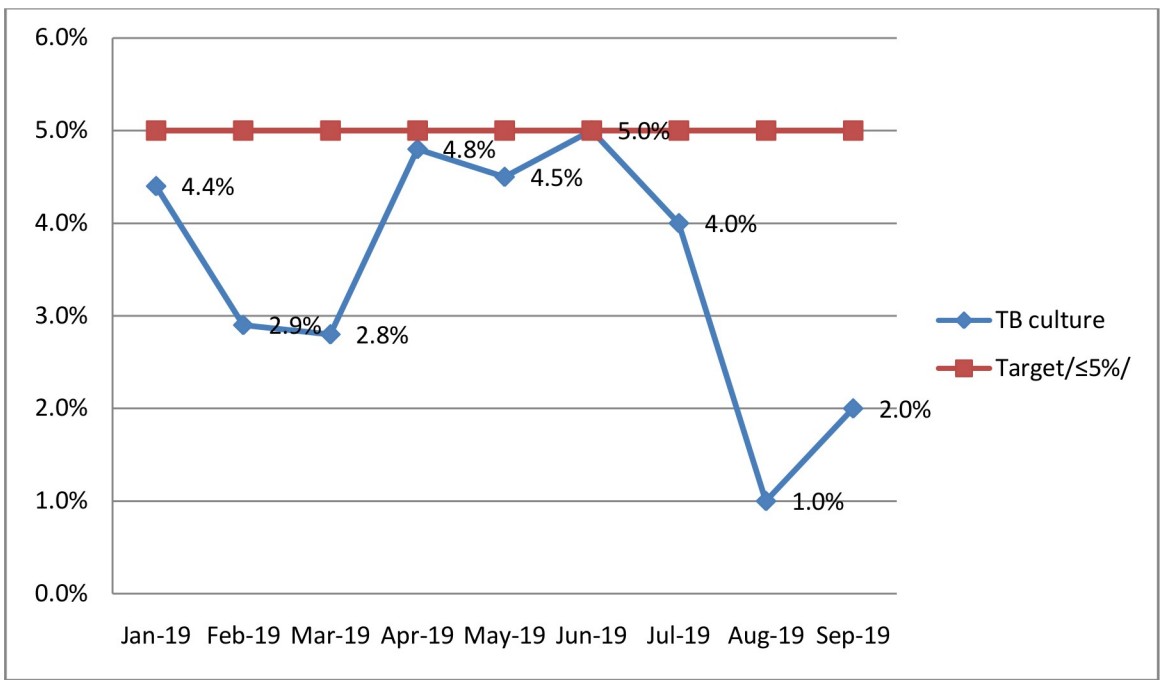

**Fig 2. Trend of TB culture contamination rates in APHI from January to September 2019.**

collect sufficient samples and process centrifugation of collected samples, and use appropriate sample collection containers in order to avoid rejection of samples referred to the institute from peripheral facilities in Amhara region since our data support that nearly half (46.1%) of the rejected specimens due to centrifugation problem and 40.9% because of use of wrong specimen containers.

Without reliable, affordable, and portable devices, regular testing remains a challenge [22]. In this study, the EID testing service was continuously interrupted for a period of over two months in addition to the interruption of liquid TB culture (MGIT method) testing. The reagent stock out was a national problem and identifying a back-up laboratory was not effective as none of the laboratories were able to obtain the reagents. Testing centers in Ghana had a shortage of EID reagents [23]. Equipment down time was also a major cause of test interruptions. Studies in Addis Ababa Ethiopia and in Malawi also revealed that the major reported factors affecting the provision of quality services were a shortage of resources and equipment failure [24, 25]. As a quality improvement plan, the laboratory could reduce interruption of tests through rapid implementation of equipment maintenance service agreements, making

**Table 5. Proportion of out of turnaround time of patient results in APHI reference laboratories, January to September 2019.**

| Tests | 2019 | | | | | | | | |
|---|---|---|---|---|---|---|---|---|---|
| | Jan | Feb | Mar | Apr | May | June | Jul | Aug | Sept |
| Viral load (%) | 15.0 | 12.8 | 0.0 | 0.0 | 18.6 | 0.0 | 0.0 | 12.6 | 81.5 |
| EID (%) | ND | 18.4 | ND | ND | 34.5 | 50.6 | 67.4 | 100.0 | 100.0 |
| TB culture (%) | 2.1 | 3.4 | 0.0 | 0.0 | 1.6 | 0.0 | 0.0 | 0.0 | 10.8 |
| GeneXpert TB (%) | 0.0 | 0.0 | 0.0 | 3.7 | 6.8 | 15.4 | 7.3 | 6.8 | 6.8 |

EID: exposed infant diagnosis; ND: not done; TB: tuberculosis.

**Table 6. Average monthly TAT of HIV and TB related laboratory tests, APHI, January to September 2019.**

| Tests | 2019 | | | | | | | | | |
|---|---|---|---|---|---|---|---|---|---|---|
| | Jan | Feb | Mar | Apr | May | June | Jul | Aug | Sept | |
| Viral load/ day | 12.7 | 13 | 7.6 | 8.9 | 21.8 | 21.9 | 19.2 | 19.1 | 24.1 | 16.5 |
| EID/day | 24 | 23.4 | ND | ND | 45.6 | 30.6 | 24.3 | 32.8 | 29.3 | 30.0 |
| TB culture/ day | 50 | 43.3 | 48.7 | 49.8 | 53.4 | 51.6 | 63 | 65.7 | 62.8 | 54.3 |
| GeneXpert TB /day | 4.0 | 3.6 | 1.0 | 2.0 | 2.9 | 1.6 | 1.6 | 2 | 2.3 | 2.3 |

EID: exposed infant diagnosis; ND: not done; TB: tuberculosis.

back up referring laboratory functional, and building maintenance workshop when there is equipment failure. In addition, the institute should establish a system for international assistance for advanced laboratory test reagents and supplies when these items are stocked out at national level.

An efficient laboratory workflow ensures that specimens received in the laboratory are tested within the established laboratory turnaround time and results are returned to health care providers and their patients on time. However, specimen backlogs can occur when there is suboptimal workflow in the laboratory network as the result of poor sample quality, reagent or consumable stock out, and equipment breakdown [26]. In this study, the TAT of viral load and EID tests were longer than target limits. About 82% of viral load tests and 100% of the EID tests had out of targeted TAT in September 2019. It was higher than a study done in Myanmar where long TAT was observed in 69% of the participants [27], and in Kenya and Malawi where EID and viral load tests lasted average TAT of 24.7 and 8 days, respectively [28, 29]. This could be due to stock-outs of reagents and maintenance issues with the automated PCR testing equipment since our data showed reagent stock out of EID in about 54 days and equipment down time for a total of 22 days in the nine months from January to September 2019. In general, the TAT for viral load and EID showed increasing delays up to the end of September 2019.

## Limitations

Percentage of incorrect laboratory test reports and notification of critical results were not evaluated since we didn't have these data during the study period. Similarly, we didn't measure IQC performance in the analytical phase of testing process.

## Conclusions

In this study, there were high rates of TB culture and viral load specimen rejection due to centrifugation problems and the use of wrong specimen containers. Equipment downtime and reagent stock out were the main causes of EID test interruption. The trend of TAT of viral load and EID tests were longer than the targeted average TAT. Hence, training of sample collectors, functional equipment maintenance systems and improved supply chain management are recommended for continuous quality improvement in addition to conducting a large scale study to identify more factors.

## Acknowledgments

The authors would like to acknowledge the Amhara Public Health Institute for cooperation to use the database and review records.

## Author Contributions

**Conceptualization:** Melashu Balew Shiferaw.

**Data curation:** Melashu Balew Shiferaw, Abay Sisay Misganaw.

**Formal analysis:** Melashu Balew Shiferaw.

**Methodology:** Melashu Balew Shiferaw, Abay Sisay Misganaw.

**Resources:** Melashu Balew Shiferaw.

**Supervision:** Melashu Balew Shiferaw, Abay Sisay Misganaw.

**Validation:** Melashu Balew Shiferaw.

**Writing – original draft:** Melashu Balew Shiferaw.

**Writing – review & editing:** Melashu Balew Shiferaw, Abay Sisay Misganaw.

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
