## [Decision Letter · Decision Letter 0]

10 Dec 2019

PONE-D-19-30524

Evaluation of continuous quality improvement of tuberculosis and HIV diagnostic services in Amhara public health institute, Ethiopia

PLOS ONE

Dear Mr Shiferaw,

Thank you for submitting your manuscript to PLOS ONE. After careful consideration, we feel that it has merit but does not fully meet PLOS ONE’s publication criteria as it currently stands. Therefore, we invite you to submit a revised version of the manuscript that addresses the points raised during the review process.

We would appreciate receiving your revised manuscript by Jan 24 2020 11:59PM. To enhance the reproducibility of your results, we recommend that if applicable you deposit your laboratory protocols in protocols.io, where a protocol can be assigned its own identifier (DOI) such that it can be cited independently in the future. For instructions see: http://journals.plos.org/plosone/s/submission-guidelines#loc-laboratory-protocols

We look forward to receiving your revised manuscript.

Kind regards,

Evelyn Byrd Quinlivan, MD

Academic Editor

PLOS ONE

Journal Requirements:

Please ensure that your manuscript meets PLOS ONE's style requirements, including those for file naming. The PLOS ONE style templates can be found at http://www.plosone.org/attachments/PLOSOne_formatting_sample_main_body.pdf and http://www.plosone.org/attachments/PLOSOne_formatting_sample_title_authors_affiliations.pdf

2. In the ethics statement in the manuscript and in the online submission form, please provide additional information about the patient records/samples used in your retrospective study.

Specifically, please ensure that you have discussed whether all data/samples were fully anonymized before you accessed them and/or whether the IRB or ethics committee waived the requirement for informed consent.

If patients provided informed written consent to have data/samples from their medical records used in research, please include this information.

Reviewers' comments:

Reviewer's Responses to Questions

**Comments to the Author**

1. Is the manuscript technically sound, and do the data support the conclusions?

Reviewer #1: Yes

Reviewer #2: Yes

2. Has the statistical analysis been performed appropriately and rigorously? 

Reviewer #1: N/A

Reviewer #2: Yes

3. Have the authors made all data underlying the findings in their manuscript fully available?

Reviewer #1: Yes

Reviewer #2: No

4. Is the manuscript presented in an intelligible fashion and written in standard English?

Reviewer #1: Yes

Reviewer #2: Yes

5. Review Comments to the Author

Reviewer #1: Great Efforts by the authors but more work sill needed to further improve this work. Please see details bellow for authors follow-up

Abstract:

Background:

• The focus of this study is continuous quality improvement (CQI) assessment using specific laboratory indicators but information on CQI not provided.

• The established target was your laboratory specific and should be made clear as such in line 30 including specific target for each quality indicators.

• Replace long waiting time with TAT and effect same correction through out the text for uniformity and consistency of terminology.

Methods:

• Line 26 through 30 should contain information of number of facilities and level of health care if such data is available. This would help for another level of analysis to improve the work further. How was the data collected, reviewed cleaned and analyzed was not stated.

• Line 38 stated that ‘’EID had a long waiting time in September 2019’’. What specific time are you referring to?. Include the exact data for the TAT.

Introduction:

• The authors may wish to consider this statement Continuous quality improvement (CQI) is a useful objective tool to improve processes and services and linking it with line 45 through 48 and ensuring effective flow and synchronization. Pre-analytical error and Post-analytical error data are abundant and well documented and would be important to include them in the introduction.

• Line 54 replace ‘’equipment down’’ with equipment down time and do same throughout text and include internal quality control (IQC) after proficiency testing in line 55.

• Replace “has been magnified more’’ with would become magnified in line 66 and expunge such and of in line 74 and 76 respectively.

Materials and Methods:

Authors may wish to consider subheading using the following guide for easy readability:

1. Study Design and ensure the actual study design was well stated since data was retrospective

2. Setting should be well describe including longitude and latitude of the study facility

3. Data Collection

4. Quality indicators and objective of each

5. Data review, cleaning and data Analysis

6. Ethical Approval or Ethical Consideration

7. Table for quality indicator and threshold or target for the laboratory

Line 91 replace maintained with repair or fix and do same for gexpert with GeneXpert in line 93.

Please define all abbreviations such as RR, RRF, RRF and DBS in text before they can be use.

What of Internal quality control monitoring threshold or target for qualitative data such as viral load testing?

Results Section:

• You may wish to consider including characteristics of facilities and various level of health cares including ownership (private and public) in the result section. Include percentage in parenthesis for each absolute number from line 132 through 137.

• Ensure you do not repeat results in tables for example use term like about half or close to half or one-third etc.

• Between line 160 and 167 ensure that text are presented as they are arrange in heading and remove was in line 164 after rate. Replace March (2.8%) to June (5.0%) in 2019 with what you have in line 164.

• Please replaced waiting time with TAT with waiting time through out text see Line 175 through 190.

• What of GeneXpert Data?

• In line 179 replace interestingly with However and expunge was in 181 after TAT, include 2019 after May in line 181 and same in line 189 after February.

Discussion:

• Authors may wish to consider major findings summary as the first paragraph and discussion of results chronologically as presented in result section in next paragraphs thereafter

• Include period of months in line 193 and replace This finding and is in line 195 with Those findings and was.

• Where was 0.26% reported by Cao et al., in line 197 and modify sample recollection with repeat specimen collection in line 197 and any other place recollection of sample appear in text

• In line 199, include compromise before patient safety and modify resources wastes to waste resources.

• In line 203 replace sample collectors with Phlebotomy and do so through text and replace sole in line 205 with avoid and include nearly half in line 2016 before 46.1% and include specimen before containers in line 207

• In line 209 include for a period of after interrupted and remove in the study after month.

• The sentence in line 212 to 213 require modification and done after studies should be expunge

• In line 222 replace is with was and in line 223 remove and replace observed or recorded with seen and line 224 that the should be replace with where and include period after months in 227.

Conclusion:

Effect on necessary correction pointed out in discussion in conclusion include specimen in line 231 before container and time after down and phlebotomy in line 233 to replace samples collectors and include improved in line 234 after and before supply.

Tables and Figures

• Table 1 Epert TB should read GeneXpert TB and include Abbreviations Definition under table

• Table 2 inlcude n(%) for each test and include Abbreviations and Definition

• Include figure title under each Chart for Line Graph for Fig 1 and Fig 2 respectively

Finally,authors may wish to consider consulting an English expert for proof reading and Plos one Journal Editor may wish to carry our independent plagiarism check on the manuscript.

Reviewer #2: This paper describes performance measures for TB culture, TB rifampicin resistance (RR) testing using GeneXpert, HIV viral load testing, and early infant diagnosis (EID) of HIV testing. This is one of the first papers to assess quality indicators for laboratory testing in Ethiopia. The gaps and successes in laboratory testing in Ethiopia outlined in this paper offer several important contributions to the literature and can guide more effective testing processes in Ethiopia and other similar settings.

The manuscript requires re-organization and editing to clarify the results and be more accessible to non-specialists. By section, my suggestions are as follows:

Introduction

1. Lines 54-58: You mention that “recommended quality indicators for the post-analytical phase are turnaround time (TAT), the percentage of incorrect laboratory test reports, and notification of critical results,” yet you do not measure or report on incorrect laboratory results or notification of critical results. Do you have these data? If not, consider adding to the limitations.

2. Lines 59-62: Confusing sentence, consider revising

3. Lines 68-71: The statistics for MRD/RR-TB and ART use seem out of place here and hard to contextualize in terms of laboratory testing requirements. Would be more useful to know – at a national level – how many patients require each of these tests annually.

Methods

4. Line 111: Should be “Oneworld Accuracy”, please ensure correct spelling and capitalization. Also suggest citing

5. Line 93: GeneXpert spelled incorrectly.

6. Lines 89: How does the indicator “test interruption” differ from equipment downtime and stock out?

7. Line 95: Unclear to reader what “Besides, reagent stock out status was set to be zero” means.

8. Line 102: Suggest rephrasing to say, “sample received 5+ days after collection…”

9. Line 104, “old” plasma samples – please quantify what how old is “old”

10. Lines 109-114, please describe in more detail what proficiency testing entails.

u

11. Lines 112-118: How were these targets (PT 80%, contamination <5%, error <5%, TAT for each test) established?

Results

1. The presentation of results is difficult to follow. I would suggest presenting the all of results for each test separately and then moving on to the next test, keeping the same format and organization for the presentation of results of each test. For example, discuss TB culture first and talk about:

a. Pre-examination indicators:

i. total number of samples for culture TB

ii. rejection rate for culture TB and reasons for rejection

b. Examination phase indicators:

i. Test interruption for culture TB & reasons

ii. Error/contamination rates and reasons for culture TB

c. Post examination phase indicators:

i. TAT for culture TB

Then repeat this sequence of results presentation for GeneXpert, EID, and VL.

2. The inclusion of data per month complicates the results, without adding much benefit, since there does not seem to be a temporal trend in indicators. I would suggest removing references to months when specific rates were achieved. Instead, the results could be simplified by first giving the average across the entire study period and then giving the low to high range, rather than specify the value per month.

a. For example, Lines 165-167 could be simplified to “Regarding TB GeneXpert, the average error rate over the study period was XX%, with monthly error rates ranging from 0%-4%.”

3. No examination phase measures for HIV viral load testing are discussed.

4. average TAT across the study period was not specified for each test, rather, monthly rates were given, which unnecessarily complicates the results.

5. TAT for GeneXpert RR tests are not specified, other than “good”.

6. Tables 4 and 5. What is the difference between turnaround time and waiting time?

7. Table 4: I’d suggest a title that makes it clearer that the numbers presented are rates of samples that achieved TAT targets. As is, that is not clear.

Discussion

• The authors miss an opportunity to discuss recommendations to address the gaps that their data highlights. The only recommendation offered discusses the need for training to address sample rejection issues (lines 202-206). While important, this will not help with machine down, stockouts and long turnaround times – what can be done to improve indicators for these?

• Please discuss the 2 months where no EID tests were conducted – what happened to the EID samples during the period: were they sent to other labs? Were they held on to and processed once the machine was repaired and reagents replenished – if so, this could explain the increased sample volume and TAT observed in May?

• Line 221-222: “About 82% of viral load tests and 100% of the EID tests had an average TAT of 45.6 days in September 2019.”

• Suggest adding a limitations section.

General comments

1. Please define all acronyms at first use. Some that were not accurately defined include: MDR/RR-TB (line 68); IFRR/RRF (line 96); DRTB (line 98), MGIT (line 152), LJ culture (line 154)

2. Please be sure to write GeneXpert with correct capitalization pattern

6. PLOS authors have the option to publish the peer review history of their article (what does this mean?). If published, this will include your full peer review and any attached files.

Reviewer #1: No

Reviewer #2: No

---

## [Author Response · Author response to Decision Letter 0]

14 Dec 2019

'Response to Reviewers'. 

Dear editor, thank you so much for mailing the comments. We would like to thank the reviewer’s, really well appreciated critics and we learn a lot from the review, and also our work now has been improved much from their comments. The following is our point by point response:

Journal Requirements:

In the ethics statement in the manuscript and in the online submission form, please provide additional information about the patient records/samples used in your retrospective study. Specifically, please ensure that you have discussed whether all data/samples were fully anonymized before you accessed them and/or whether the IRB or ethics committee waived the requirement for informed consent. If patients provided informed written consent to have data/samples from their medical records used in research, please include this information.

Response: thank you, now detailed ethics statement has been included in the manuscript and in the online submission form. 

Reviewers' comments:

Reviewer #1: 

Abstract:

Background:

• The focus of this study is continuous quality improvement (CQI) assessment using specific laboratory indicators but information on CQI not provided.

Response: Thank you, now it has been provided in the revised manuscript

• The established target was your laboratory specific and should be made clear as such in line 30 including specific target for each quality indicators.

Response: Thank you, it has been made clear now since include specific target for each quality indicators in table 1 of the new version manuscript. But, to include in the abstract part it became more detailed and out of word limit. So, table 1 and description in the method part is enough.

• Replace long waiting time with TAT and effect same correction throughout the text for uniformity and consistency of terminology.

Response: thank you, it has been replaced throughout the text

Methods:

• Line 26 through 30 should contain information of number of facilities and level of health care if such data is available. This would help for another level of analysis to improve the work further. How was the data collected, reviewed cleaned and analyzed was not stated.

Response: thank you, now we have stated how the data was collected, reviewed cleaned and analyzed. However, information of number of facilities and level of health care data was not available to include as suggested.

• Line 38 stated that ‘’EID had a long waiting time in September 2019’’. What specific time are you referring to?. Include the exact data for the TAT.

Response: thank you, now it has been described. The specific time was 29.3 days as described in the new version manuscript. 

Introduction:

• The authors may wish to consider this statement Continuous quality improvement (CQI) is a useful objective tool to improve processes and services and linking it with line 45 through 48 and ensuring effective flow and synchronization. Pre-analytical error and Post-analytical error data are abundant and well documented and would be important to include them in the introduction.

Response: : thank you, pre-analytical error and Post-analytical error data has been included them in the introduction

• Line 54 replace ‘’equipment down’’ with equipment down time and do same throughout text and include internal quality control (IQC) after proficiency testing in line 55.

Response: thanks, now it has been corrected as the reviewer’s advice. 

• Replace “has been magnified more’’ with would become magnified in line 66 and expunge such and of in line 74 and 76 respectively.

Response: thanks, now it has been corrected as the reviewer’s advice.

Materials and Methods:

Authors may wish to consider subheading using the following guide for easy readability:

1. Study Design and ensure the actual study design was well stated since data was retrospective

Response: thanks, now it has been corrected

2. Setting should be well describe including longitude and latitude of the study facility

Response: Now it has been well described as shown in the setting part of the method section

3. Data Collection

Response: thanks, now it has been revised

4. Quality indicators and objective of each

Response: thanks, now it has been revised and we include it together with data collection part

5. Data review, cleaning and data Analysis

Response: thanks, now it has been revised

6. Ethical Approval or Ethical Consideration

Response: thanks, now it has been revised

7. Table for quality indicator and threshold or target for the laboratory

Response: Thank you, now the table has been included as the reviewer’s advice. 

Line 91 replace maintained with repair or fix and do same for gexpert with GeneXpert in line 93.

Response: thanks, now it has been corrected

Please define all abbreviations such as RR, RRF, RRF and DBS in text before they can be use.

Response: thanks, now it has been revised

What of Internal quality control monitoring threshold or target for qualitative data such as viral load testing?

Response: Thanks, it is 100%. It is described in table 1 of the new version manuscript. 

Results Section:

• You may wish to consider including characteristics of facilities and various level of health cares including ownership (private and public) in the result section. Include percentage in parenthesis for each absolute number from lie 132 through 137.

Response: Thanks, now we have included percentage in parenthesis for each absolute number from line 132 through 137. However, we couldn’t get data for characteristics of health facilities to include as the reviewers recommendation. 

• Ensure you do not repeat results in tables for example use term like about half or close to half or one-third etc.

Response: Thank you for the comment; it has been revised according to the reviewers advice. 

• Between line 160 and 167 ensure that text are presented as they are arrange in heading and remove was in line 164 after rate. Replace March (2.8%) to June (5.0%) in 2019 with what you have in line 164.

Response: Thank you, now corrections have been made based on the reviewer comments. 

• Please replaced waiting time with TAT with waiting time throughout text see Line 175 through 190.

Response: Thank you, now waiting time has been replaced with TAT in the revised version manuscript.

• What of GeneXpert Data?

Response: Now, we specified in the result section under TAT in addition to table 5 

• In line 179 replace interestingly with However and expunge was in 181 after TAT, include 2019 after May in line 181 and same in line 189 after February.

Response: Thank you, now it has been corrected

Discussion:

• Authors may wish to consider major findings summary as the first paragraph and discussion of results chronologically as presented in result section in next paragraphs thereafter

Response: Thanks, now we add summary of results in the first paragraph of the discussion section as the reviewer’s advice.

• Include period of months in line 193 and replace This finding and is in line 195 with Those findings and was.

Response: Now, it has been corrected

• Where was 0.26% reported by Cao et al., in line 197 and modify sample recollection with repeat specimen collection in line 197 and any other place recollection of sample appear in text

Response: Thanks, it was in Huston, United States. And recollection has been modified based on the reviewer’s advice.

• In line 199, include compromise before patient safety and modify resources wastes to waste resources.

Response: Thank you very much; it really helps us to improve our paper. And it has been modified accordingly. 

• In line 203 replace sample collectors with Phlebotomy and do so through text and replace sole in line 205 with avoid and include nearly half in line 2016 before 46.1% and include specimen before containers in line 207

Response: Thank you; all recommendations have been corrected in the new version manuscript except Phlebotomy that it better to use as it’s, because the term phlebotomy more related to blood collections, but in our study, we have more than blood sample like sputum.

• In line 209 include for a period of after interrupted and remove in the study after month.

Response: Thank you; now it has been modified 

• The sentence in line 212 to 213 require modification and done after studies should be expunge

Response: Thank you; now it has been modified

• In line 222 replace is with was and in line 223 remove and replace observed or recorded with seen and line 224 that the should be replace with where and include period after months in 227.

Response: Now, it has been corrected

Conclusion:

Effect on necessary correction pointed out in discussion in conclusion include specimen in line 231 before container and time after down and phlebotomy in line 233 to replace samples collectors and include improved in line 234 after and before supply.

Response: Now, it has been corrected

Tables and Figures

• Table 1 Epert TB should read GeneXpert TB and include Abbreviations Definition under table

Response: Now, it has been corrected

• Table 2 inlcude n(%) for each test and include Abbreviations and Definition

Response: Response: Now, it has been corrected

• Include figure title under each Chart for Line Graph for Fig 1 and Fig 2 respectively

Response: Thank you for the comment. The journal requirement describes to provide figures in separate and title to include in the main manuscript. That is why we didn’t put under each figure. The editor may put it under the graph during publication. 

Reviewer #2: 

Introduction

1. Lines 54-58: You mention that “recommended quality indicators for the post-analytical phase are turnaround time (TAT), the percentage of incorrect laboratory test reports, and notification of critical results,” yet you do not measure or report on incorrect laboratory results or notification of critical results. Do you have these data? If not, consider adding to the limitations.

Response: Thank you, comment accepted and it has been described as limitation at the end of the discussion part

2. Lines 59-62: Confusing sentence, consider revising

Response: Thank you, now it has been revised to avoid the confusion.

3. Lines 68-71: The statistics for MRD/RR-TB and ART use seem out of place here and hard to contextualize in terms of laboratory testing requirements. Would be more useful to know – at a national level – how many patients require each of these tests annually.

Response: Thank you. The statistics described was to give background information how much of the patients suffer from HIV and TB related health problems at national level including our setting. All of the listed statistics require quality assured laboratory results. So, we believe this information is important and has not been deleted. 

Methods

4. Line 111: Should be “Oneworld Accuracy”, please ensure correct spelling and capitalization. Also suggest citing

Response: Thank you, now it has been corrected and cited.

5. Line 93: GeneXpert spelled incorrectly.

Response: Thank you, now it has been corrected

6. Lines 89: How does the indicator “test interruption” differ from equipment downtime and stock out?

Response: Thank you, it differs because the lab uses back up referring laboratories to avoid test interruption when there is either equipment downtime or stock out. 

7. Line 95: Unclear to reader what “Besides, reagent stock out status was set to be zero” means.

Response: now it has been modified. It means no reagent stock out was set as a quality indicator by APHI.

8. Line 102: Suggest rephrasing to say, “sample received 5+ days after collection…”

Response: Thank you, now it has been rephrased as the reviewer’s advice

9. Line 104, “old” plasma samples – please quantify what how old is “old”

Response: Thanks, it was said old plasma samples if it was delayed >5 days after collection when transported to APHI at 2-8oC. Now it has been included in the new manuscript.

10. Lines 109-114, please describe in more detail what proficiency testing entails.

Response: Now it has been more entailed and included in the new version manuscript. 

11. Lines 112-118: How were these targets (PT 80%, contamination <5%, error <5%, TAT for each test) established?

Response: Thanks, now it has been included in table 1 of the new version manuscript

Results

1. The presentation of results is difficult to follow. I would suggest presenting the all of results for each test separately and then moving on to the next test, keeping the same format and organization for the presentation of results of each test. For example, discuss TB culture first and talk about:

a. Pre-examination indicators:

i. total number of samples for culture TB

ii. rejection rate for culture TB and reasons for rejection

b. Examination phase indicators:

i. Test interruption for culture TB & reasons

ii. Error/contamination rates and reasons for culture TB

c. Post examination phase indicators:

i. TAT for culture TB

Then repeat this sequence of results presentation for GeneXpert, EID, and VL.

Response: Thank you for the comments. But when we tried to rearrange based on the suggested sequence of results, it became more complicated and difficult to describe repeated quality indicators. So, we prefer the current sequence of results presentation. 

2. The inclusion of data per month complicates the results, without adding much benefit, since there does not seem to be a temporal trend in indicators. I would suggest removing references to months when specific rates were achieved. Instead, the results could be simplified by first giving the average across the entire study period and then giving the low to high range, rather than specify the value per month.

a. For example, Lines 165-167 could be simplified to “Regarding TB GeneXpert, the average error rate over the study period was XX%, with monthly error rates ranging from 0%-4%.”

Response: now it has been simplified based on the reviewer’s advice.

3. No examination phase measures for HIV viral load testing are discussed.

Response: Thank you, it has been already discussed in the result section under proficienc testing performance stating “All of the tests participated in the PT program qualified the minimum requirement that the passing mark was 80%. Specifically, viral load, EID and GeneXpert had 100% performance as evaluated in April 2019”. But IQC was not discussed since we didn’t evaluate it and we put as limitation in the new version manuscript.

4. average TAT across the study period was not specified for each test, rather, monthly rates were given, which unnecessarily complicates the results.

Response: Thank you, now it has been included in the result section and in table 5.

5. TAT for GeneXpert RR tests are not specified, other than “good”.

Response: Now, it has been specified in the new version manuscript

6. Tables 4 and 5. What is the difference between turnaround time and waiting time?

Response: Thank you so much, comments help us to improve the paper. Table 4 describes the proportion of out of TAT patient results in APHI, and Table 5 describes the average TAT of tests. In the new version manuscript, the title of table 4 has been modified as “Proportion of out of turnaround time of patient results in APHI reference laboratories, January to September 2019”

7. Table 4: I’d suggest a title that makes it clearer that the numbers presented are rates of samples that achieved TAT targets. As is, that is not clear.

Response: Thank you the title now has been cleared. Yes, rates of samples that achieved TAT targets

Discussion

• The authors miss an opportunity to discuss recommendations to address the gaps that their data highlights. The only recommendation offered discusses the need for training to address sample rejection issues (lines 202-206). While important, this will not help with machine down, stockouts and long turnaround times – what can be done to improve indicators for these?

Response: Thank you so much, very important view and help to improve our work. Now, recommendations have been included to address the gaps highlighted. 

• Please discuss the 2 months where no EID tests were conducted – what happened to the EID samples during the period: were they sent to other labs? Were they held on to and processed once the machine was repaired and reagents replenished – if so, this could explain the increased sample volume and TAT observed in May?

Response: Thank you, the 2 months EID interruption was due to the national stock out occurred in Ethiopia that all of the labs in Ethiopia were not able to test EID. It was impossible to send samples to other laboratories. In addition, equipment was failed in April for seven days as explained in the result section. Maintenance was done after that. No, these issues have been explained in the discussion section. 

• Line 221-222: “About 82% of viral load tests and 100% of the EID tests had an average TAT of 45.6 days in September 2019.”

Response: Thank you so much, now it has been corrected as “About 82% of viral load tests and 100% of the EID tests had out of targeted TAT in September 2019”.

• Suggest adding a limitations section.

Response: thanks, now limitation section has been included in the new version manuscript. 

General comments

1. Please define all acronyms at first use. Some that were not accurately defined include: MDR/RR-TB (line 68); IFRR/RRF (line 96); DRTB (line 98), MGIT (line 152), LJ culture (line 154)

Response: thanks, now it has been revised

2. Please be sure to write GeneXpert with correct capitalization pattern

Response: thanks, now it has been revised

---

## [Decision Letter · Decision Letter 1]

4 Feb 2020

PONE-D-19-30524R1

Evaluation of continuous quality improvement of tuberculosis and HIV diagnostic services in Amhara public health institute, Ethiopia

PLOS ONE

Dear Mr Shiferaw,

Thank you for submitting your manuscript to PLOS ONE. After careful consideration, we feel that it has merit but does not fully meet PLOS ONE’s publication criteria as it currently stands. Therefore, we invite you to submit a revised version of the manuscript that addresses the points raised during the review process.

The revisions address the reviewers comments regarding content but the manuscript needs further editing for language.  Attached is a copy of the submitted revision, highlights many of the grammatical issues that need correction. 

We would appreciate receiving your revised manuscript by Mar 20 2020 11:59PM. To enhance the reproducibility of your results, we recommend that if applicable you deposit your laboratory protocols in protocols.io, where a protocol can be assigned its own identifier (DOI) such that it can be cited independently in the future. For instructions see: http://journals.plos.org/plosone/s/submission-guidelines#loc-laboratory-protocols

We look forward to receiving your revised manuscript.

Kind regards,

Evelyn Byrd Quinlivan, MD

Academic Editor

PLOS ONE

Additional Editor Comments (if provided):

The manuscript addresses the reviewers comments. The manuscript should be heavily edited for grammar prior to publication.

Reviewers' comments:

Reviewer's Responses to Questions

**Comments to the Author**

1. If the authors have adequately addressed your comments raised in a previous round of review and you feel that this manuscript is now acceptable for publication, you may indicate that here to bypass the “Comments to the Author” section, enter your conflict of interest statement in the “Confidential to Editor” section, and submit your "Accept" recommendation.

Reviewer #1: All comments have been addressed

Reviewer #2: All comments have been addressed

2. Is the manuscript technically sound, and do the data support the conclusions?

Reviewer #1: Yes

Reviewer #2: Yes

3. Has the statistical analysis been performed appropriately and rigorously? 

Reviewer #1: N/A

Reviewer #2: Yes

4. Have the authors made all data underlying the findings in their manuscript fully available?

Reviewer #1: Yes

Reviewer #2: Yes

5. Is the manuscript presented in an intelligible fashion and written in standard English?

Reviewer #1: Yes

Reviewer #2: Yes

6. Review Comments to the Author

Reviewer #1: After reviewing the responses of the authors during the second stage of this review process i have the following comments that would guide the Academic editor on the final decision on the work.

1, The authors have painstakingly addressed all issues and comments raised during the first review outcomes

2. All modification and correction have been effected thereby improving the work further for publication

3. Authors have also reviewed each comments and concerns raised and provided useful explanation on various modification/correction made or why modification may be possible is few cases.

4. The Academic Editor may wish to do independent Plagiarism check before acceptance and final publication

Reviewer #2: The revised manuscript is much clearer, with reviewer comments adequately addressed. I do recommend a final review by a native English speaker to improve the fluency of the writing.

7. PLOS authors have the option to publish the peer review history of their article (what does this mean?). If published, this will include your full peer review and any attached files.

Reviewer #1: No

Reviewer #2: No

---

## [Author Response · Author response to Decision Letter 1]

5 Feb 2020

Dear editor, thanks for providing us the comments, and also we thank the reviewers. Accordingly, we have addressed requested revisions that the manuscript has been further edited for language based on the recommendations. See the new version manuscript and the tracked change manuscript submitted.

---

## [Editor Report · Decision Letter 2]

3 Mar 2020

Evaluation of continuous quality improvement of tuberculosis and HIV diagnostic services in Amhara public health institute, Ethiopia

PONE-D-19-30524R2

Dear Dr. Shiferaw,

We are pleased to inform you that your manuscript has been judged scientifically suitable for publication and will be formally accepted for publication once it complies with all outstanding technical requirements.

With kind regards,

Evelyn Byrd Quinlivan, MD

Academic Editor

PLOS ONE
---

## [Editor Report · Acceptance letter]

5 Mar 2020

PONE-D-19-30524R2 

Evaluation of continuous quality improvement of tuberculosis and HIV diagnostic services in Amhara public health institute, Ethiopia 

Dear Dr. Shiferaw:

I am pleased to inform you that your manuscript has been deemed suitable for publication in PLOS ONE. Congratulations! Your manuscript is now with our production department. 

With kind regards,

on behalf of

Dr. Evelyn Byrd Quinlivan 

Academic Editor

PLOS ONE